# Circulatory Failure among Hospitalizations for Heatstroke in the United States

**DOI:** 10.3390/medicines7060032

**Published:** 2020-06-14

**Authors:** Tarun Bathini, Charat Thongprayoon, Tananchai Petnak, Api Chewcharat, Wisit Cheungpasitporn, Boonphiphop Boonpheng, Ronpichai Chokesuwattanaskul, Narut Prasitlumkum, Saraschandra Vallabhajosyula, Wisit Kaewput

**Affiliations:** 1Department of Internal Medicine, University of Arizona, Tucson, AZ 85721, USA; 2Division of Nephrology and Hypertension, Department of Medicine, Mayo Clinic, Rochester, MN 55905, USA; chewcharat.Api@mayo.edu; 3Division of Pulmonary and Critical Care Medicine, Faculty of Medicine Ramathibodi Hospital, Mahidol University, Bangkok 10400, Thailand; petnak@yahoo.com; 4Division of Nephrology, Department of Internal Medicine, University of Mississippi Medical Center, Jackson, MS 39216, USA; 5Department of Medicine, University of California, Los Angeles, CA 90095, USA; boonpipop.b@gmail.com; 6Faculty of Medicine, King Chulalongkorn Memorial Hospital, Chulalongkorn University, Bangkok 10330, Thailand; dr_ronpichai_c@yahoo.com; 7Department of Medicine, University of Hawaii, Honolulu, HI 96822, USA; narutpra@hawaii.edu; 8Department of Cardiovascular Medicine, Mayo Clinic, Rochester, MN 55905, USA; Vallabhajosyula.Saraschandra@mayo.edu; 9Department of Military and Community Medicine, Phramongkutklao College of Medicine, Bangkok 10400, Thailand

**Keywords:** outcomes, hospitalization, heatstroke, heat stroke, resource utilization, hospitalized patients, circulatory failure, cardiology, internal medicine, medicine

## Abstract

**Background:** This study aimed to assess the risk factors and the association of circulatory failure with treatments, complications, outcomes, and resource utilization in hospitalized patients for heatstroke in the United States. **Methods:** Hospitalized patients with a principal diagnosis of heatstroke were identified in the National Inpatient Sample dataset from the years 2003 to 2014. Circulatory failure, defined as any type of shock or hypotension, was identified using hospital diagnosis codes. Clinical characteristics, in-hospital treatment, complications, outcomes, and resource utilization between patients with and without circulatory failure were compared. **Results:** A total of 3372 hospital admissions primarily for heatstroke were included in the study. Of these, circulatory failure occurred in 393 (12%) admissions. Circulatory failure was more commonly found in obese patients, but less common in older patients aged ≥60 years. The need for mechanical ventilation, blood transfusion, and renal replacement therapy were higher in patients with circulatory failure. Hyperkalemia, hypocalcemia, metabolic acidosis, metabolic alkalosis, sepsis, ventricular arrhythmia or cardiac arrest, renal failure, respiratory failure, liver failure, neurological failure, and hematologic failure were associated with circulatory failure. The in-hospital mortality was 7.1-times higher in patients with circulatory failure. The length of hospital stay and hospitalization costs were higher when circulatory failure occurred while in the hospital. **Conclusions:** Approximately one out of nine heatstroke patients developed circulatory failure during hospitalization. Circulatory failure was associated with various complications, higher mortality, and increased resource utilizations.

## 1. Introduction

Among all heat-related illness, heatstroke is the most severe spectrum of diseases defined by a condition in which the core body temperature exceeds 40 °C following a tremendous environmental heat load that overrides the body’s heat dissipation efforts [1]; it is accompanied by clinical central nervous system dysfunction and/or other organ failures [2]. Heatstroke can be categorized based on its cause as either classic or exertional heatstroke [1,3]. Classic heatstroke is commonly found among the elderly whose thermoregulation mechanism is compromised. It is caused by an imbalance between the absorption of environmental heat and heat dissipation. By contrast, exertional heatstroke generally occurs among healthy young people who have engaged in strenuous physical activity during periods of high temperature and humidity [4,5]. This heavy activity causes excessive heat production, which overwhelms the heat-loss mechanism [6].

Approximately, heatstroke in the United States (U.S.) results in 4100 emergency department visits per year, with most occurring during the summer and requiring hospitalization [7]. The mortality rate ranges between 3–7% and is expected to rise in coming years due to climate change [8,9,10]. One of the fatal complications of heatstroke is circulatory failure resulting from prolonged hyperthermia, which leads to physiological alterations and the dysregulation of inflammatory reactions. Several mechanisms have been proposed as the pathogenesis of circulatory failure in heatstroke, including hypovolemia, cardiac dysfunction, maldistribution of blood volume, and peripheral vascular failure [11]. In addition, circulatory failure was previously reported as a prognostic factor [9,10,12,13]. Therefore, the early identification of patients with risk factors for circulatory failure, and prompt treatment among them, might result in better outcomes. However, knowledge of the risk factors and the impact of circulatory failure among patients with heatstroke remains limited.

This study aimed to assess the risk factors and the association of circulatory failure with treatments, complications, outcomes, and resource use in patients hospitalized for heatstroke in the U.S.

## 2. Materials and Methods

### 2.1. Data Source

This cohort study was conducted using the 2003–2014 National Inpatient Sample (NIS) database. The NIS is the largest all-payer inpatient database in the United States. A discharge data set from a 20% stratified sample of hospitals in United States with the patient encounter-level information, which includes principal and secondary diagnosis codes as well as procedure codes, are recorded in the NIS. Sample weight is used to generate national estimates for hospitalization nationwide. The approval from an institutional review board was exempted as the information was obtained from a de-identified public database.

### 2.2. Study Population

All patients who were admitted in hospitals from 2003 to 2014 with a principal diagnosis of heatstroke, based on International Classification of Diseases, Ninth Revision, Clinical Modification (ICD-9 CM) diagnosis code of 992.0, were included. Heatstroke patients were categorized based on the presence of circulatory failure during hospitalization. Circulatory failure, which was defined as any type of shock or hypotension, was identified using ICD-9 diagnosis of 785.5 (shock without mention of trauma), 785.50 (shock, unspecified), 785.59 (other shock without trauma, including hypovolemic shock), 785.51 (cardiogenic shock), 785.52 (septic shock), 458.8 (other specified hypotension), 458.9 (hypotension, unspecified), and 796.3 (nonspecific low blood pressure reading).

### 2.3. Data Collection

Patient characteristics, treatments, complications, and outcomes during hospitalization were identified using ICD-9 codes, as previously described ([14], Table S1). Patient characteristics included age, sex, race, smoking, alcohol drinking, obesity, diabetes mellitus, hypertension, hypothyroidism, chronic kidney disease, coronary artery disease, congestive heart failure, and atrial flutter/fibrillation. Treatments included invasive mechanical ventilation, blood component transfusion, and renal replacement therapy. Complications and outcomes included electrolyte derangements (hyponatremia, hypernatremia, hypokalemia, hyperkalemia, hypocalcemia, hypercalcemia, metabolic acidosis, metabolic alkalosis), rhabdomyolysis, sepsis, gastrointestinal bleeding, ventricular arrhythmia or cardiac arrest, end-organ failure (renal failure, respiratory failure, liver failure, neurological failure, hematological failure), and in-hospital mortality. Resource utilization included length of hospital stay and hospitalization cost. 

### 2.4. Statistical Analysis

The total number of heatstroke patients was estimated using discharge-level weights provided by the Healthcare Cost and Utilization Project (HCUP). Continuous variables were reported as mean ± standard deviation and were compared using Student’s *t*-test. Categorical variables were reported as counts with percentage and were compared using Chi-squared test. Multivariable logistic regression with forward stepwise selection was performed to assess if there were clinical characteristics independently associated with circulatory failure. The association of circulatory failure with treatments, complications, and outcomes was assessed using logistic regression analysis, and with length of hospital stay and hospitalization cost using linear regression analysis, with pre-specified adjustment for age, sex, smoking, alcohol drinking, and comorbidities. A two-tailed *p*-value of less than 0.05 was considered statistically significant. All analyses were two-tailed. Statistical significance was achieved when *p*-value < 0.05. SPSS statistical software (version 22.0, IBM Corporation, Armonk, NY, USA) was used for all analyses.

## 3. Results

### 3.1. Incidence of and Risk Factors for Circulatory Failure in Hospitalized Heatstroke Patients

A total of 3372 hospitalized heatstroke patients were included in analysis. Of these, 393 (12%) developed circulatory failure in hospital. Table 1 compares clinical characteristics, in-hospital treatments, complications, outcomes, and resource utilization between patients with and without circulatory failure. In multivariable analysis (Table 2), obesity was independently associated with increased odds of circulatory failure, while age ≥ 60 was associated with decreased odds of circulatory failure. 

### 3.2. The Association of Circulatory Failure with In-Hospital Treatments, Complication, and Outcomes

In terms of in-hospital treatments, patients with circulatory failure had higher odds of receiving invasive mechanical ventilation (OR 3.96; *p <* 0.001), blood component transfusion (OR 4.60; *p <* 0.001), and renal replacement therapy (OR 3.70; *p <* 0.001) than patients without circulatory failure. In terms of in-hospital complications and outcomes, circulatory failure was significantly associated with hyperkalemia (OR 2.66; *p <* 0.001), hypocalcemia (OR 2.77; *p <* 0.001), metabolic acidosis (OR 2.90; *p <* 0.001), metabolic alkalosis (OR 3.75; *p* = 0.001), sepsis (OR 3.93; *p <* 0.001), gastrointestinal bleeding (OR 3.43; *p* = 0.006), ventricular arrhythmia/cardiac arrest (OR 4.45; *p <* 0.001), renal failure (OR 2.03; *p <* 0.001), respiratory failure (OR 3.64; *p <* 0.001), liver failure (OR 3.27; *p <* 0.001), neurological failure (OR 1.91; *p <* 0.001), and hematological failure (OR 2.70; *p <* 0.001). Patients with circulatory failure had 7.1-times higher odds of in-hospital mortality than patients without circulatory failure (*p* < 0.001). 

### 3.3. Impact of Circulatory Failure on Resource Utilization

Compared with patients without circulatory failure, the mean length of hospital stay increased by 2.2 days (*p <* 0.001), and the mean hospitalization cost increased by $33,102 (*p <* 0.001) in patients with circulatory failure (Table 3).

## 4. Discussion

This is the largest cohort study demonstrating the impact of circulatory failure among heatstroke patients in the US. We found that 12% of hospitalized heatstroke patients had circulatory failure. The in-hospital mortality rate among hospitalized heatstroke patients with circulatory failure was 19.1%. In addition, heatstroke patients with circulatory failure were at great risk for complications, including metabolic ventricular arrhythmia, cardiac arrest, sepsis, gastrointestinal hemorrhage, metabolic abnormalities, and other end-organ failures. Further, patients had a higher risk of invasive mechanical ventilation, blood component transfusion, and renal replacement therapy. Obesity was associated with higher odds of circulatory failure, while advanced age was associated with lower odds of circulatory failure.

Circulatory failure is common among heatstroke patients. Our study demonstrated that patients who experience circulatory failure have a 7-fold higher mortality rate than those who do not. Circulatory failure is a well-known and significant factor for predicting deaths among the critically ill [15]. Several previous studies have confirmed the impact of circulatory failure on death rates among heatstroke patients [9,10,12,13]. Furthermore, circulatory failure not only affects mortality rates, it also results in various consequent complications. The occurrence of circulatory failure might reflect an uncompensated state in severe cases. Circulatory failure can result in a reduction of tissue perfusion and oxygen delivery, causing a variety of complications. The disproportion between metabolic demand and oxygen delivery during systemic inflammatory response in severe heatstroke may lead to type IV respiratory failure requiring mechanical ventilation [16]. In addition, hypovolemia coexisting with myocardial dysfunction in severe cases may enhance the deterioration of renal blood flow, leading to acute kidney injury [17,18]. 

In terms of the gastrointestinal tract, hemodynamic change, hypotension, and vasopressor use may reduce microcirculation flow of the gastrointestinal tract. During circulatory failure, there is systemic vasoconstriction of the arteriolar resistance vessels throughout the body, especially mesenteric vasoconstriction, to increase the circulating blood volume into the systemic circulation [19]. In addition to activation of the sympathetic nervous system, the angiotensin II receptors of the mesenteric vascular resistance smooth muscle have a 5-fold affinity for angiotensin II. Thus gastrointestinal perfusion is often compromised early relative to other vascular beds during circulatory failure [20]. The hypoperfusion along the gastrointestinal tract can lead to the dysfunction of several protective mechanisms causing stress-related mucosal ulcers and gastrointestinal hemorrhage [21]. In addition, an increase of catecholamine tone during cardiovascular collapse, combined with extensive muscle injury-induced electrolyte abnormalities, might also enhance ventricular arrhythmia and cardiac arrest [22,23]. These complications result in longer hospital stays and require more invasive treatments, which may, in turn, increase the risk of hospital-acquired infection and sepsis. Finally, complications from circulatory failure, such as multiorgan failures, electrolyte abnormalities, and cardiac arrhythmia, may also result in cardiac dysfunction. This vicious cycle cannot be explicitly concluded as the cause and effect of these complications with circulatory failure. 

In this study, we demonstrated that obesity was associated with higher odds of circulatory failure. Obese patients tend to have higher heat production accompanied by insulator characteristics of adipose tissue. Higher sweat rates may be required to help dissipate the excess heat, as in previous reports [24]. These higher sweat rates lead to water and salt loss. This loss, combined with systemic inflammatory response syndrome from prolonged hyperthermia, results in cardiovascular collapse and circulatory failure [25,26]. Circulatory failure in heatstroke may be induced by several mechanisms, including hypovolemia caused by severe dehydration, heat stress-induced myocardial dysfunction, and the pooling of blood volume at the periphery [11]. In addition to common mechanisms, exertional heatstroke seems to have a higher risk of hemodynamic compromise due to right ventricular dysfunction, which was not found in non-exertional heatstroke [18,27]. An animal study demonstrated metabolic dysfunction in the myocardium following exertional heatstroke [28]. These mechanisms might explain why younger patients who tend to suffer from exertional heatstroke are at higher risk of developing circulatory failure, while elderly heatstroke patients may be susceptible to non-exertional heatstroke, which is less likely to develop circulatory failure than exertional heatstroke.

This study has some limitations. First, the study may not detect patients whose condition is complicated with circulatory failure at the scene, since the NIS database is hospital-based. We may underestimate the impact of circulatory failure if patients are not admitted, or if they fully recover from cardiovascular collapse before admission. Moreover, we could not demonstrate information prior to hospital admission, such as the onset of heatstroke, and the long-term sequelae of circulatory failure among heatstroke patients. Given the structure of the NIS database, we could only show short-term complications and outcomes during their hospital stays. Finally, the study could not demonstrate differences in outcomes regarding the type of heatstroke.

## 5. Conclusions

In conclusion, approximately one out of nine heatstroke patients developed circulatory failure during hospitalization. Circulatory failure is associated with various complications, higher mortality rates, and increased resource use. Early detection, prompt treatment, and close monitoring of these complications might lower mortality and decrease resource use.

## Figures and Tables

**Table 1 medicines-07-00032-t001:** Clinical characteristics, in-hospital treatments, complications, outcomes, and resource utilization in heatstroke patients.

	Total	Circulatory Failure	No Circulatory Failure	*p*-Value
Clinical characteristics				
N (%)	3372	393 (11.7)	2979 (88.3)	
Age (years)	55 ± 22	51 ± 22	55 ± 22	<0.001
<20	218 (6.5)	38 (9.7)	180 (6.0)	
20–39	654 (19.4)	68 (17.3)	586 (19.7)	
40–59	1034 (30.7)	147 (37.4)	887 (29.8)	
60–79	900 (26.7)	93 (23.7)	807 (27.1)	
≥80	564 (16.7)	47 (12.0)	517 (17.4)	
Male	2478 (73.6)	94 (23.9)	795 (26.7)	0.23
Race				0.04
Caucasian	1883 (55.8)	218 (55.5)	1665 (55.9)	
African American	496 (14.7)	53 (13.5)	443 (14.9)	
Hispanic	428 (12.7)	66 (16.8)	362 (12.2)	
Other	565 (16.8)	56 (14.2)	509 (17.1)	
Smoking	604 (17.9)	58 (14.8)	546 (18.3)	0.08
Alcohol drinking	270 (8.0)	38 (9.7)	232 (7.8)	0.20
Obesity	233 (6.9)	44 (11.2)	189 (6.3)	<0.001
Diabetes Mellitus	562 (16.7)	57 (14.5)	505 (17.0)	0.22
Hypertension	1255 (37.2)	131 (33.3)	1124 (37.7)	0.09
Hypothyroidism	196 (5.8)	25 (6.4)	171 (5.7)	0.62
Chronic kidney disease	201 (6.0)	28 (7.1)	173 (5.8)	0.30
Coronary artery disease	389 (11.5)	40 (10.2)	349 (11.7)	0.37
Congestive heart failure	216 (6.4)	24 (6.1)	192 (6.4)	0.80
Atrial flutter/fibrillation	251 (7.4)	30 (7.6)	221 (7.4)	0.88
Treatment				
Invasive mechanical ventilation	686 (20.3)	178 (45.3)	508 (17.1)	<0.001
Blood component transfusion	169 (5.0)	62 (15.8)	107 (3.6)	<0.001
Renal replacement therapy	75 (2.2)	26 (6.6)	49 (1.6)	<0.001
Complication and outcomes				
Hyponatremia	293 (8.7)	41 (10.4)	252 (8.5)	0.19
Hypernatremia	183 (5.4)	25 (6.4)	158 (5.3)	0.38
Hypokalemia	500 (14.8)	47 (12.0)	453 (15.2)	0.09
Hyperkalemia	134 (4.0)	34 (8.7)	100 (3.4)	<0.001
Hypocalcemia	73 (2.2)	19 (4.8)	54 (1.8)	<0.001
Hypercalcemia	37 (1.1)	6 (1.5)	31 (1.0)	0.39
Metabolic acidosis	472 (14.0)	116 (29.5)	356 (12.0)	<0.001
Metabolic alkalosis	29 (0.9)	10 (2.5)	19 (0.6)	<0.001
Rhabdomyolysis	1049 (31.1)	143 (36.4)	906 (30.4)	0.02
Gastrointestinal bleeding	54 (1.6)	21 (5.3)	33 (1.1)	<0.001
Sepsis	149 (4.4)	49 (12.5)	100 (3.4)	<0.001
Ventricular arrhythmia/cardiac arrest	95 (2.8)	33 (8.4)	62 (2.1)	<0.001
Renal failure	1226 (36.4)	203 (51.7)	1023 (34.3)	<0.001
Respiratory failure	550 (16.3)	147 (37.4)	403 (13.5)	<0.001
Liver failure	196 (5.8)	58 (14.8)	138 (4.6)	<0.001
Neurological failure	651 (19.3)	118 (30.0)	533 (17.9)	<0.001
Hematological failure	449 (13.3)	108 (27.5)	341 (11.4)	<0.001
In-hospital mortality	168 (5.0)	75 (19.1)	93 (3.1)	<0.001
Resource utilization				
Length of hospital stay (days)	4.3 ± 7.0	6.4 ± 10.6	4.0 ± 6.3	<0.001
Hospitalization cost ($)	35,335 ± 72,085	66,999 ± 120,863	31,140 ± 61,671	<0.001

Continuous variables are reported as mean ± standard deviation, categorical variables as counts (percentages).

**Table 2 medicines-07-00032-t002:** Univariable and multivariable analysis assessing factors associated with circulatory failure in heatstroke patients.

Variables	Univariable Analysis		Multivariable Analysis	
	Crude Odds Ratio (95%CI)	*p*-Value	Adjusted Odds Ratio (95%CI)	*p*-Value
Age (years)				
<20	1.27 (0.86–1.88)	0.23	1.32 (0.89–1.95)	0.16
20–39	0.70 (0.52–0.95)	0.02	0.72 (0.53–0.97)	0.03
40–59	1 (reference)		1 (reference)	
60–79	0.70 (0.53–0.92)	0.01	0.71 (0.54–0.94)	0.02
≥80	0.55 (0.39–0.78)	0.001	0.58 (0.41–0.81)	0.002
Male	1.16 (0.91–1.48)	0.24		
Race				
Caucasian	1 (reference)			
African American	0.91 (0.67–1.26)	0.58		
Hispanic	1.39 (1.03–1.88)	0.03		
Other	0.84 (0.62–1.15)	0.27		
Smoking	0.77 (0.58–1.04)	0.08		
Alcohol drinking	1.27 (0.88–1.82)	0.20		
Obesity	1.86 (1.32–2.63)	<0.001	1.76 (1.24–2.50)	0.002
Diabetes Mellitus	0.83 (0.62–1.12)	0.22		
Hypertension	0.83 (0.66–1.03)	0.09		
Hypothyroidism	1.12 (0.72–1.72)	0.62		
Chronic kidney disease	1.24 (0.82–1.88)	0.30		
Coronary artery disease	0.42 (0.21–0.83)	0.01		
Congestive heart failure	0.82 (0.41–1.62)	0.57		
Atrial flutter/fibrillation	1.04 (0.58–1.87)	0.89		

**Table 3 medicines-07-00032-t003:** The association of circulatory failure with in-hospital treatment, complications, outcomes, and resource utilization in heatstroke patients.

	Univariable Analysis		Multivariable Analysis	
	Crude Odds Ratio (95% CI)	*p*-Value	Adjusted Odds Ratio * (95% CI)	*p*-Value
Treatments				
Invasive mechanical ventilation	4.03 (3.23–5.02)	<0.001	3.96 (3.11–5.03)	<0.001
Blood component transfusion	5.03 (3.60–7.01)	<0.001	4.60 (3.25–6.53)	<0.001
Renal replacement therapy	4.24 (2.60–6.90)	<0.001	3.70 (2.15–6.35)	<0.001
Complications and outcomes				
Hyponatremia	1.26 (0.89–1.79)	0.19	1.28 (0.89–1.82)	0.18
Hypernatremia	1.21 (0.79–1.88)	0.39	1.19 (0.76–1.86)	0.44
Hypokalemia	0.76 (0.55–1.04)	0.09	0.79 (0.57–1.09)	0.14
Hyperkalemia	2.73 (1.82–4.09)	<0.001	2.66 (1.76–4.04)	<0.001
Hypocalcemia	2.75 (1.61–4.69)	<0.001	2.77 (1.60–4.79)	<0.001
Hypercalcemia	1.47 (0.61–3.56)	0.39	1.55 (0.62–3.86)	0.35
Metabolic acidosis	3.09 (2.42–3.94)	<0.001	2.90 (2.25–3.73)	<0.001
Metabolic alkalosis	4.07 (1.88–8.81)	<0.001	3.75 (1.68–8.40)	0.001
Rhabdomyolysis	1.31 (1.05–1.63)	0.02	1.25 (0.99–1.58)	0.06
Sepsis	4.10 (2.86–5.88)	<0.001	3.93 (2.70–5.73)	<0.001
Gastrointestinal bleeding	5.04 (2.89–8.80)	<0.001	3.43 (1.42–8.25)	0.006
Ventricular arrhythmia/cardiac arrest	4.31 (2.79–6.67)	<0.001	4.45 (2.83–7.00)	<0.001
Renal failure	2.04 (1.65–2.53)	<0.001	2.03 (1.62–2.54)	<0.001
Respiratory failure	3.82 (3.04–4.81)	<0.001	3.64 (2.86–4.64)	<0.001
Liver failure	3.56 (2.57–4.94)	<0.001	3.27 (2.30–4.65)	<0.001
Neurological failure	1.97 (1.56–2.49)	<0.001	1.91 (1.50–2.42)	<0.001
Hematological failure	2.93 (2.29–3.76)	<0.001	2.70 (2.07–3.52)	<0.001
In-hospital mortality	7.30 (5.28–10.11)	<0.001	7.09 (5.04–9.96)	<0.001
	Coefficient (95% CI)	*p*-value	Adjusted coefficient (95% CI)	*p*-value
Resource utilization				
Length of hospital stay (days)	2.4 (1.6–3.1)	<0.001	2.2 (1.4–2.9)	<0.001
Hospitalization cost ($)	35,859 (28,330–43,388)	<0.001	33,102 (25,596–40,609)	<0.001

* Adjusted for age, sex, race, smoking, alcohol drinking, obesity, diabetes mellitus, hypertension, hypothyroidism, and chronic kidney disease, chronic ischemic heart disease, congestive heart failure, and atrial flutter/fibrillation.

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
