# Peer review of "Circulatory Failure among Hospitalizations for Heatstroke in the United States"

_medicines, 2020, doi:10.3390/medicines7060032_

Round 1

Reviewer 1 Report

Retrospective analysis of a cohort of 3372 patients with heat stroke in the United States. Appropriate methodology and presentation of findings. Although it is more than obvious that a patient with heat stroke will have a worse evolution if he or she presents associated circulatory failure, studies like this one are necessary to corroborate that hypothesis.

Author Response

Response to Reviewer #1

Retrospective analysis of a cohort of 3372 patients with heat stroke in the United States. Appropriate methodology and presentation of findings. Although it is more than obvious that a patient with heat stroke will have a worse evolution if he or she presents associated circulatory failure, studies like this one are necessary to corroborate that hypothesis.

Response: We thank you for reviewing our manuscript and for your critical evaluation. The reviewers’ inputs are extremely helpful. We believe as a result of this review; our study would have more value for the readers. We revised the manuscript based on the reviewer’s suggestions.

We greatly appreciated the editor and reviewer’s time and comments to improve our manuscript.

Reviewer 2 Report

This is a nicely written paper describing a retrospective analysis of risk factors associated with circulatory failure during heat stroke. I only have a few comments for the authors to consider.

  • It would be of value to the reader to provide (even if its conjecture) the actual mechanism which causes the circulatory failure. From what I understand, the GIT vasoconstricts to support the peripheral vasodilation. But the overall systemic circulatory failure occurs once the a GIT can no longer vasoconstrict enough and shock results. If correct, what is the signal that causes the GIT to lose tone and vasodilate?   Commenting on this would bring some biological chronology to the paper.
  • I suspect that the time the patient has been hyperthermic before arriving to the hospital is highly correlated with whether or not they experience circulatory failure. In other words, if the person has only been hyperthermic for a short amount of time prior to admission, they are ostensibly less likely to develop a circulatory failure than the person who had been hypothermic for a longer time. Can the authors comment on this?
  • Adipose tissue is relatively metabolically inactive, but it does act as a good insulator. This is the reason why obese patients accumulate more heat and thus need to sweat more.    

Author Response

Response to Reviewer#2

This is a nicely written paper describing a retrospective analysis of risk factors associated with circulatory failure during heat stroke. I only have a few comments for the authors to consider

Response: We thank you for reviewing our manuscript and for your critical evaluation.

Comment #1

It would be of value to the reader to provide (even if its conjecture) the actual mechanism which causes the circulatory failure. From what I understand, the GIT vasoconstricts to support the peripheral vasodilation. But the overall systemic circulatory failure occurs once the a GIT can no longer vasoconstrict enough and shock results. If correct, what is the signal that causes the GIT to lose tone and vasodilate?   Commenting on this would bring some biological chronology to the paper.

Response: Reviewer raises very important point to improve our manuscript. We agree with this important point and thus we additionally added the discussion on mesenteric responses to circulatory failure in our discussion as the reviewer’s suggestion. While during septic shock may be associated with increased (vasodilatation) or decreased (vasoconstriction) mesenteric blood flow but is characterized by increased oxygen consumption, exceeding the capability of mesenteric oxygen delivery, Circulatory failure in heat stroke (hypovolemic or cardiogenic shock) often results in decreased perfusion pressure, prompting selective vasoconstriction of the mesenteric arterioles to maintain perfusion pressure of the vital organs, here at the selective expense of the mesenteric organs. The following text has been added as the reviewer’s suggestion:

“In terms of the gastrointestinal tract, hemodynamic change, hypotension, and vasopressor use may reduce microcirculation flow of the gastrointestinal tract. During circulatory failure, there is systemic vasoconstriction of the arteriolar resistance vessels throughout the body, especially mesenteric vasoconstriction, to increase the circulating blood volume into the systemic circulation (19). In addition to activation of the sympathetic nervous system, the angiotensin II receptors of the mesenteric vascular resistance smooth muscle have a 5-fold affinity for angiotensin II. Thus gastrointestinal perfusion is often compromised early relative to other vascular beds during circulatory failure (20). The hypoperfusion along the gastrointestinal tract can lead to the dysfunction of several protective mechanisms causing stress-related mucosal ulcers and gastrointestinal hemorrhage (21).”

Comment #2

I suspect that the time the patient has been hyperthermic before arriving to the hospital is highly correlated with whether or not they experience circulatory failure. In other words, if the person has only been hyperthermic for a short amount of time prior to admission, they are ostensibly less likely to develop a circulatory failure than the person who had been hypothermic for a longer time. Can the authors comment on this?

Response: We agreed with the reviewer that the time from heatstroke to hospital arrival might affect patient outcomes, including circulatory failure. However, the NIS database did not have information before hospital admission.  The following statements have been added to acknowledge this limitation

Moreover, we could not demonstrate information prior to hospital admission, such as the onset of heatstroke, and the long-term sequelae of circulatory failure among heatstroke patients.

Comment #3

Adipose tissue is relatively metabolically inactive, but it does act as a good insulator. This is the reason why obese patients accumulate more heat and thus need to sweat more.   

Response: We agree with the reviewer’s important point. The following statements have been added to discussion.

Obese patients tend to have higher heat production accompanied by insulator characteristics of adipose tissue. Higher sweat rates may require to help dissipate the excess heat, as the previous report [21]. These higher sweat rates lead to water and salt loss. This loss, combined with systemic inflammatory response syndrome from prolonged hyperthermia, results in cardiovascular collapse and circulatory failure [22,23].

We greatly appreciated the editor and reviewer’s time and comments to improve our manuscript.

Reviewer 3 Report

Overall, this is manuscript is a valuable contribution to the heat stroke literature.  In general, it is well written and I believe the analytic and statistical methods are valid. 

Specific Comments: 

  • 95% of the methods section is exact word-for-word copy of the methods from a previous paper written by the same authors. https://www.mdpi.com/2077-0383/9/5/1357/htm. "Acute Myocardial infarction among hospitalization for heat stroke in the United States."  I am not accusing the authors of plagiarism, as that was a different study, with different objectives but with the same database. I think the authors should acknowledge the methodology as described in the previous paper and should put quotes around the text with the reference.  That would be appropriate from an ethical standpoint.
  • I find myself puzzled by the effects of age. Perhaps the authors were too.  In general, as I am sure they know, in large population studies in heat stroke from heat waves, age has a negative effect on outcomes.  The authors discuss the possibility (in the discussion) that it could be due to the fact that the older patients may not have had exertional heat stroke…..which may be more likely to lead to circulatory failure.  Our laboratory has studied this in mice and indeed we have only seen the development of heart failure like conditions in an exertional heat stroke model PMID: 32026469, though this has not been studied extensively in passive heat stroke.  I would like to ask the authors to rewrite that short section of the discussion to make it clearer to the reader what you are hypothesizing a specific reason why. It is not an unreasonable hypothesis, but in its current form it is not exactly clear. 
  • In table one, I think the authors could be clearer to include an up or down arrow or a +/- sign for the variables that are higher or lower in the circulatory failure group. For example it is very hard for the reader to decipher it from the numbers printed without doing the proportions themselves. 
  • Why not include an adjusted odds ratio for multivariate analysis for coronary artery disease in Table 2? i.e. Coronary artery disease    42 (0.21-0.83)   P< 0.01.  This is important.  Was that calculated in your other paper?….you could just reference it.  I think it important.  Reference your other paper when it is appropriate and don't worry about being accused of self-citation.  It is appropriate here. 
  • Overall a nice contribution.  

Author Response

Response to Reviewer#3

Overall, this is manuscript is a valuable contribution to the heat stroke literature.  In general, it is well written and I believe the analytic and statistical methods are valid. 

Response: We thank you for reviewing our manuscript and for your critical evaluation.

Comment #1

95% of the methods section is exact word-for-word copy of the methods from a previous paper written by the same authors. https://www.mdpi.com/2077-0383/9/5/1357/htm. "Acute Myocardial infarction among hospitalization for heat stroke in the United States."  I am not accusing the authors of plagiarism, as that was a different study, with different objectives but with the same database. I think the authors should acknowledge the methodology as described in the previous paper and should put quotes around the text with the reference.  That would be appropriate from an ethical standpoint.

Response: We agree with the reviewer. We have cited this reference as suggested.

  1. Bathini T, Thongprayoon C, Chewcharat A, Petnak T, Cheungpasitporn W, Boonpheng B, et al. Acute Myocardial Infarction among Hospitalizations for Heat Stroke in the United States. J Clin Med. 2020;9(5)

Comment #2

I find myself puzzled by the effects of age. Perhaps the authors were too.  In general, as I am sure they know, in large population studies in heat stroke from heat waves, age has a negative effect on outcomes.  The authors discuss the possibility (in the discussion) that it could be due to the fact that the older patients may not have had exertional heat stroke…..which may be more likely to lead to circulatory failure.  Our laboratory has studied this in mice and indeed we have only seen the development of heart failure like conditions in an exertional heat stroke model PMID: 32026469, though this has not been studied extensively in passive heat stroke.  I would like to ask the authors to rewrite that short section of the discussion to make it clearer to the reader what you are hypothesizing a specific reason why. It is not an unreasonable hypothesis, but in its current form it is not exactly clear.

Response: We appreciate the reviewer’s important input. We agree and the following statements have been added to discussion. We found the suggested reference (PMID: 32026469) very helpful and have utilized them as new reference (28) Laitano O, Garcia CK, Mattingly AJ, Robinson GP, Murray KO, King MA, et al. Delayed metabolic dysfunction in myocardium following exertional heat stroke in mice. J Physiol. 2020;598(5):967-85. The following text in bold has been added to the discussion.

“Circulatory failure in heatstroke may be induced by several mechanisms, including hypovolemia caused by severe dehydration, heat stress-induced myocardial dysfunction, and the pooling of blood volume at the periphery (11). In addition to common mechanisms, exertional heatstroke seems to have a higher risk of hemodynamic compromise due to right ventricular dysfunction, which was not found in non-exertional heatstroke (18, 27). An animal study demonstrated metabolic dysfunction in the myocardium following exertional heatstroke (28). These mechanisms might explain why younger patients who tend to suffer from exertional heatstroke are at higher risk of developing circulatory failure, while elderly heatstroke patients may be susceptible to non-exertional heatstroke, which is less likely to develop circulatory failure than exertional heatstroke.”

  1. Laitano O, Garcia CK, Mattingly AJ, Robinson GP, Murray KO, King MA, et al. Delayed metabolic dysfunction in myocardium following exertional heat stroke in mice. J Physiol. 2020;598(5):967-85.

Comment #3

In table one, I think the authors could be clearer to include an up or down arrow or a +/- sign for the variables that are higher or lower in the circulatory failure group. For example it is very hard for the reader to decipher it from the numbers printed without doing the proportions themselves.

Response: We agree with the reviewer. We highlighted significant p-value<0.05 in Table 1.

Comment #4

Why not include an adjusted odds ratio for multivariate analysis for coronary artery disease in Table 2? i.e. Coronary artery disease  0.42 (0.21-0.83)   P< 0.01.  This is important.  Was that calculated in your other paper?….you could just reference it.  I think it important.  Reference your other paper when it is appropriate and don't worry about being accused of self-citation.  It is appropriate here.

Response: We agree with the reviewer. We performed multivariable logistic regression with forward stepwise selection to identify clinical characteristics associated with circulatory failure. History of coronary artery disease was associated with circulatory failure with odds ratio of 0.42 (95 % CI 0.21-0.83) in univariable analysis, and 1.01 (95% CI 0.70-1.45) in multivariable analysis. In table 2, we only reported odds ratio of variables that were significantly associated with circulatory failure in multivariable analysis, (p<0.05)

Comment #5

Overall a nice contribution.

Response: We greatly appreciated the editor and reviewer’s time and comments to improve our manuscript.

This manuscript is a resubmission of an earlier submission. The following is a list of the peer review reports and author responses from that submission.

Round 1

Reviewer 1 Report

The manuscript "Circulatory failure among hospitalizations for heat stroke in the United States" is a report on 3372 hospitalizations with an ICD code of heat stroke. The question is which patients will have circulatory failure, and what the circulatory failure is associated with (question may have been two-sided: what are the initial predictors and what are the consequences; this is however not mentioned). The authors think that they can identify "circulatory failure " by gathering several ICD codes for shock (shock without mention for trauma, shock unspecified, septic shock, other specified hypotension, and even nonspecific low blood pressure reading). The authors also think that it does not matter whether these ICD codes are given at discharge or at admission or somewhere in between. There are always problems with validating the results from existing databases dependent on interpretation of different people. What is lacking is a pulse and blood pressure reading at admission, temperature reading at admission, all items that you would like to know when you ask a question about which patients will have circulatory failure after admission.

Reviewer 2 Report

First of all I would like to commend the authors' manuscript. It is well written in a concise manner and easy to read.

The overall presentation of the data is sound and I only have a minor concern. The data stems from a stratified sample taken from national registry data and is based in grouping by ICD Codes. With this approach, there is a risk for falsely attributing clinical presentation with ICD codes. Coding might influenced by the fincancial merit of some codes vs others, regardless of clinical presentation. This is not a fault by the others. However, it might be prudent to point out the limitations of studies conducted based on retrospective matching of ICD-Codes with other clinical findings.

The abbreviation HCUP (p 3, l 106) is not introduced.